# Metagenome-wide association of gut microbiome features for schizophrenia

Feng Zhu [1,19,20], Yanmei Ju[2,3,4,5,19,20], Wei Wang[6,7,8,19,20], Qi Wang [2,5,19,20], Ruijin Guo[2,3,4,9,19,20], Qingyan Ma[6,7,8], Qiang Sun[2,10], Yajuan Fan[6,7,8], Yuying Xie[11], Zai Yang[6,7,8], Zhuye Jie[2,3,4], Binbin Zhao[6,7,8], Liang Xiao [2,3,12], Lin Yang[6,7,8], Tao Zhang [2,3,13], Junqin Feng[6,7,8], Liyang Guo[6,7,8], Xiaoyan He[6,7,8], Yunchun Chen[6,7,8], Ce Chen[6,7,8], Chengge Gao[6,7,8], Xun Xu [2,3], Huanming Yang[2,14], Jian Wang[2,14], Yonghui Dang[15], Lise Madsen[2,16,17], Susanne Brix [2,18], Karsten Kristiansen [2,17,20✉], Huijue Jia [2,3,4,9,20✉] & Xiancang Ma [6,7,8,20✉]

Evidence is mounting that the gut-brain axis plays an important role in mental diseases fueling mechanistic investigations to provide a basis for future targeted interventions. However, shotgun metagenomic data from treatment-naïve patients are scarce hampering comprehensive analyses of the complex interaction between the gut microbiota and the brain. Here we explore the fecal microbiome based on 90 medication-free schizophrenia patients and 81 controls and identify a microbial species classifier distinguishing patients from controls with an area under the receiver operating characteristic curve (AUC) of 0.896, and replicate the microbiome-based disease classifier in 45 patients and 45 controls (AUC = 0.765). Functional potentials associated with schizophrenia include differences in short-chain fatty acids synthesis, tryptophan metabolism, and synthesis/degradation of neurotransmitters. Transplantation of a schizophrenia-enriched bacterium, *Streptococcus vestibularis*, appear to induces deficits in social behaviors, and alters neurotransmitter levels in peripheral tissues in recipient mice. Our findings provide new leads for further investigations in cohort studies and animal models.

[1] Center for Translational Medicine, The First Affiliated Hospital of Xi'an Jiaotong University, 277 Yanta West Road, Xi'an 710061, China. [2] BGI-Shenzhen, Shenzhen 518083, China. [3] China National Genebank, Shenzhen 518120, China. [4] Shenzhen Key Laboratory of Human Commensal Microorganisms and Health Research, BGI-Shenzhen, Shenzhen 518083, China. [5] BGI Education Center, University of Chinese Academy of Sciences, Shenzhen 518083, China. [6] Department of Psychiatry, The First Affiliated Hospital of Xi'an Jiaotong University, 277 Yanta West Road, Xi'an 710061, China. [7] Center for Brain Science, The First Affiliated Hospital of Xi'an Jiaotong University, 277 Yanta West Road, Xi'an 710061, China. [8] Clinical Research Center for Psychiatric Medicine of Shaanxi Province, The First Affiliated Hospital of Xi'an Jiaotong University, 277 Yanta West Road, Xi'an 710061, China. [9] Macau University of Science and Technology, Taipa, Macau 999078, China. [10] Department of Statistical Sciences, University of Toronto, Toronto, Canada. [11] Department of Statistics and Probability, Michigan State University, East Lansing, USA. [12] Shenzhen Engineering Laboratory of Detection and Intervention of Human Intestinal Microbiome, BGI-Shenzhen, Shenzhen 518083, China. [13] Shenzhen Key Laboratory of Cognition and Gene Research, BGI-Shenzhen, Shenzhen 518083, China. [14] James D. Watson Institute of Genome Sciences, Hangzhou 310058, China. [15] School of Forensic Medicine, Xi'an Jiaotong University, 76 Yanta West Road, Xi'an 710061, China. [16] Institute of Marine Research (IMR), P.O. Box 7800 , 5020 Bergen, Norway. [17] Laboratory of Genomics and Molecular Biomedicine, Department of Biology, University of Copenhagen, Universitetsparken 13, 2100 Copenhagen, Denmark. [18] Department of Biotechnology and Biomedicine, Technical University of Denmark, 2800 Kgs, Lyngby, Denmark. [19] These authors contributed equally: Feng Zhu, Yanmei Ju, Wei Wang, Qi Wang, Ruijin Guo. [20] These authors jointly supervised: Karsten Kristiansen, Huijue Jia, Xiancang Ma. ✉email: kk@bio.ku.dk; jiahuijue@genomics.cn; maxiancang@163.com

Schizophrenia is a severe psychiatric disorder associated with hallucinations, delusions, and thought disorders perturbing perception and social interaction[1]. The etiology of schizophrenia is not elucidated, but assumed to be multifactorial involving genetic and environmental factors. Abnormalities of neurotransmitter systems have been extensively studied especially focusing on aberration of signaling involving dopamine, glutamate, and γ-aminobutyric acid (GABA)[2–4]. Accumulating evidence indicates that schizophrenia may be a systemic disorder with neuropsychiatric conditions in addition to psychosis[5]. Furthermore, the importance of inflammation[6] and the involvement of the gastrointestinal system[7] in schizophrenia have received attention.

The gut microbiota is reported to play an important role in neurogenerative processes, and perturbation of the microbiota and microbial products have been demonstrated to affect behavior[8–10]. Changes in the gut microbiota have been associated with neurological[11] and neurodevelopmental disorders[12,13], and recently also by an independent study of schizophrenia[14]. It was recently reported that fecal transfer of the gut microbiota from patients with schizophrenia induces schizophrenia-associated behaviors in germ-free recipient mice accompanied with altered levels of glutamate, glutamine, and GABA in the hippocampus[14]. However, the identity and functionality of the specific bacteria responsible for mediating changes in the behavior of recipient mice are unknown[15,16]. Thus, the composition and functional capacity of the gut microbiota in relation to schizophrenia need to be systematically examined. Taxonomic and functional profiling of the gut microbiota is required for functional understanding of the gut microbiota[17]. Metagenomic shotgun sequencing combined with bioinformatics tools enables better characterization of the microbiota[18], including a more accurate prediction of biological features of the microbes and their potential influence on host physiology[19].

Here, we report on a metagenome-wide association study (MWAS) using 171 samples (90 cases and 81 controls) and validate the results by analyzing additional 90 samples (45 cases and 45 controls). The functional changes characterizing the schizophrenia gut microbiota are determined using pathway/module analysis based on the Kyoto Encyclopedia of Genes and Genomes (KEGG) and a recently developed gut–brain module (GBM) analysis of fecal metagenomes[20]. The possible role of one particular schizophrenia-enriched gut bacterial species, *Streptococcus* (S.) *vestibularis*, is explored by transplanting this bacterium into the gut of the mice with antibiotic-induced microbiota depletion and observing its effects on animal behavior and brain neurochemicals.

## Results

### The gut microbiota profile of schizophrenic patients.
We carried out shotgun sequencing on fecal samples from 90 medication-free patients and 81 healthy controls (for demographic and clinical characteristics see Supplementary Data 1–3) and obtained an average of 11.46 gigabases (Gb) sequence data per sample and mapped the high-quality reads onto a comprehensive reference gene catalog of 11.4 million genes[21] (Supplementary Data 4).

The gut microbiota in schizophrenic patients showed greater α diversity at the genus level ($P = 0.027$, Wilcoxon rank-sum test), higher β diversity at the genus level ($P < 0.001$, Wilcoxon rank-sum test) and microbial gene level ($P < 0.001$, Wilcoxon rank-sum test) and comprised more genes compared with healthy controls (Supplementary Fig. 1). Out of a total of 360 metagenomic operational taxonomic units (mOTUs)[22], 83 mOTUs showed significant differences in relative abundance between patients and

controls ($P < 0.05$ and false discovery rate (FDR) = 0.136, Wilcoxon rank sum test and Storey's FDR method; Supplementary Data 5a). After adjusting for BMI, age, sex, and diet, these 83 mOTUs were still significant (Supplementary Data 5a). The gut microbiota in schizophrenic patients harbored many facultative anaerobes such as *Lactobacillus fermentum, Enterococcus faecium, Alkaliphilus oremlandii*, and *Cronobacter sakazakii/turicensis*, which are rare in a healthy gut. Additionally, bacteria that are often present in the oral cavity, such as *Veillonella atypica, Veillonella dispar, Bifidobacterium dentium, Dialister invisus, Lactobacillus oris*, and *Streptococcus salivarius* were more abundant in patients with schizophrenia than in healthy controls, indicating a close association between the oral and the gut microbiota in schizophrenia.

We then constructed a mOTU network to depict the co-occurrence correlation between the schizophrenia-associated gut bacteria (Fig. 1). Schizophrenia-enriched mOTUs were more interconnected than control-enriched mOTUs (Spearman's correlation coefficient $<-0.3$ or $>0.3$, $P < 0.05$). The mOTU species from the genera *Streptococcus* and *Veillonella* showed positive cross-correlations. Moreover, the majority of the species in these two clusters of correlated mOTUs originated from the oral cavity, again pointing to the relation between oral resident bacteria and gut bacteria, suggesting that oral resident bacteria in a synergistic manner may colonize the gut in schizophrenic patients (Fig. 1 and Supplementary Data 5a).

Functional modules and pathways enriched in the gut microbiota of patients relative to controls were analyzed using the KEGG database (Supplementary Data 6). The relative enrichment of 579 KEGG modules and 323 KEGG pathways varied significantly between the two groups. Schizophrenia-depleted microbial functional modules included pectin degradation, lipopolysaccharide biosynthesis, autoinducer-2 (AI-2) transport system, glutamate/aspartate transport system, beta-carotene biosynthesis, whereas schizophrenia-enriched functional modules included methanogenesis, the gamma-aminobutyrate (GABA) shunt, and transport system of manganese, zinc, and iron (Supplementary Data 6).

### Neuroactive potential of schizophrenia-related bacteria.
We next compared the altered microbial neuroactive potential of the gut microbiota of schizophrenic patients with the controls at the species level using the method reported by Valles-Colomer et al.[20]. We mapped the metagenomic data of the 171 samples to a genome database including the 42 microbial species that were detected based on the 83 schizophrenia-associated mOTUs using PanPhlan[23] and calculated the prevalence of species-level microbes. We then determined whether the abundance of 56 previously reported gut-brain modules (GBMs)[20], present in each microbial species, varied significantly between schizophrenic patients and controls. The GBM set of each microbial species was obtained by cross-checking GBM-related genes and the species gene repertoires (Supplementary Data 7a). The frequency of the occurrence of each GBM within each species was compared between patients and controls using a Chi-squared test (Supplementary Data 7b). Schizophrenia-associated GBMs included short-chain fatty acid synthesis (acetate, propionate, butyrate, and isovaleric acid), tryptophan metabolism, and the synthesis of several neurotransmitters, such as glutamate, GABA, and nitric oxide (Fig. 2).

We chose to validate the presence of the GBM associated with tryptophan metabolism in schizophrenia, as tryptophan metabolism is modulated by the gut microbiota and implicated in schizophrenia pathogenesis[24,25]. Hence, serum tryptophan metabolites were measured in patients and controls and correlated

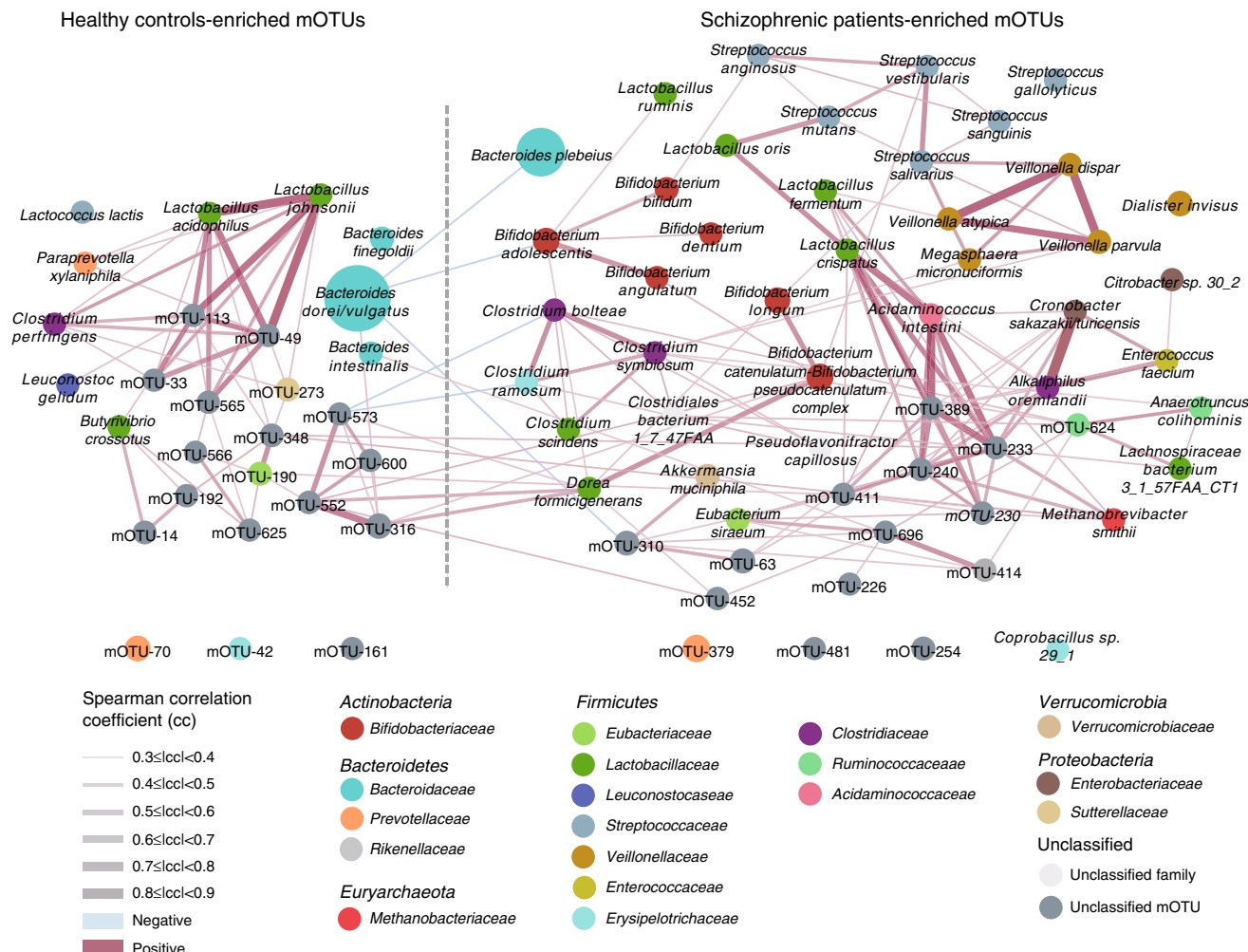

**Fig. 1 Network of mOTUs differentially enriched in healthy controls and schizophrenic patients.** Node sizes reflect the mean abundance of significant mOTUs. mOTUs annotated to species are colored according to family (Red edges, Spearman's rank correlation coefficient > 0.3, $P < 0.05$; blue edges, Spearman's rank correlation coefficient <−0.3, $P < 0.05$;). See detailed statistical data in supplementary Source Data file.

with the presence of tryptophan modules in the gut microbiota. In agreement with the higher abundance of tryptophan metabolisms related GBMs, we observed lower serum tryptophan levels and higher kynurenic acid (KYNA) levels in schizophrenic patients(Supplementary Fig. 2a, c). Moreover, serum tryptophan levels were negatively correlated with the abundances of 38 bacterial species enriched in schizophrenic patients and positively correlated with 6 bacterial species enriched in controls (Supplementary Fig. 2d). Similarly, serum KYNA levels were positively correlated with 10 schizophrenia-enriched bacterial species and negatively correlated with 3 control-enriched bacterial species (Supplementary Fig. 2d). Thus, an altered gut microbiota may be associated with changes in serum levels of tryptophan and KYNA in schizophrenia.

**Gut microbial species characteristic of schizophrenia.** To identify novel gut bacterial species associated with schizophrenia and evaluate their diagnostic values, we first constructed a set of random forest disease classifiers based on gut mOTUs. We performed a five-fold cross-validation procedure ten times on 90 patients and 81 controls. Twenty six gut mOTUs reached the lowest classifier error in the random forest cross validation, and the area under the receiver operating characteristic curve (AUC) of the model was 0.896 (Fig. 3a, b). This microbial based classifier was not significant influenced by age, gender, BMI, and diet style.

(Supplementary Data 8). This discriminatory model was validated on an additional validation cohort consisting 45 patients taking antipsychotics and 45 controls (Supplementary Data 9). The model still distinguished patients from controls with an AUC of 0.765. Among the 26 mOTUs included in the classifier, 11 bacterial species with taxonomic identity were significantly enriched in schizophrenia, namely *Akkermansia muciniphila*, *Bacteroides plebeius*, *Veillonella parvula*, *Clostridium symbiosum*, *Eubacterium siraeum*, *Cronobacter sakazakii/turicensis*, *S. vestibularis*, *Alkaliphilus oremlandii*, *Enterococcus faecium*, *Bifidobacterium longum*, and *Bifidobacterium adolescentis*. Some of these microbial species were significantly associated with symptom severity, cognitive performance, and diagnosis (Fig. 3c).

We next performed metagenomic analysis on the fecal samples from 38 of the 90 patients after 3-months of treatment (27 with risperidone and 11 with other antipsychotics, shown in Supplementary Data 1). The psychotic symptoms and cognitive impairment improved greatly along with treatment (Supplementary Fig. 3). However, only approximately half of mOTUs that distinguished SCZ patients from controls returned to the levels in controls after treatment (Fig. 3d). As the sample size of the follow-up patients was smaller, the statistical significance threshold was increased from 0.05 to 0.1. Of the 26 identified microbial species, 20 species remained significantly changed between 81 controls and 38 baseline patients ($P < 0.1$, FDR = 0.44, Benjamini

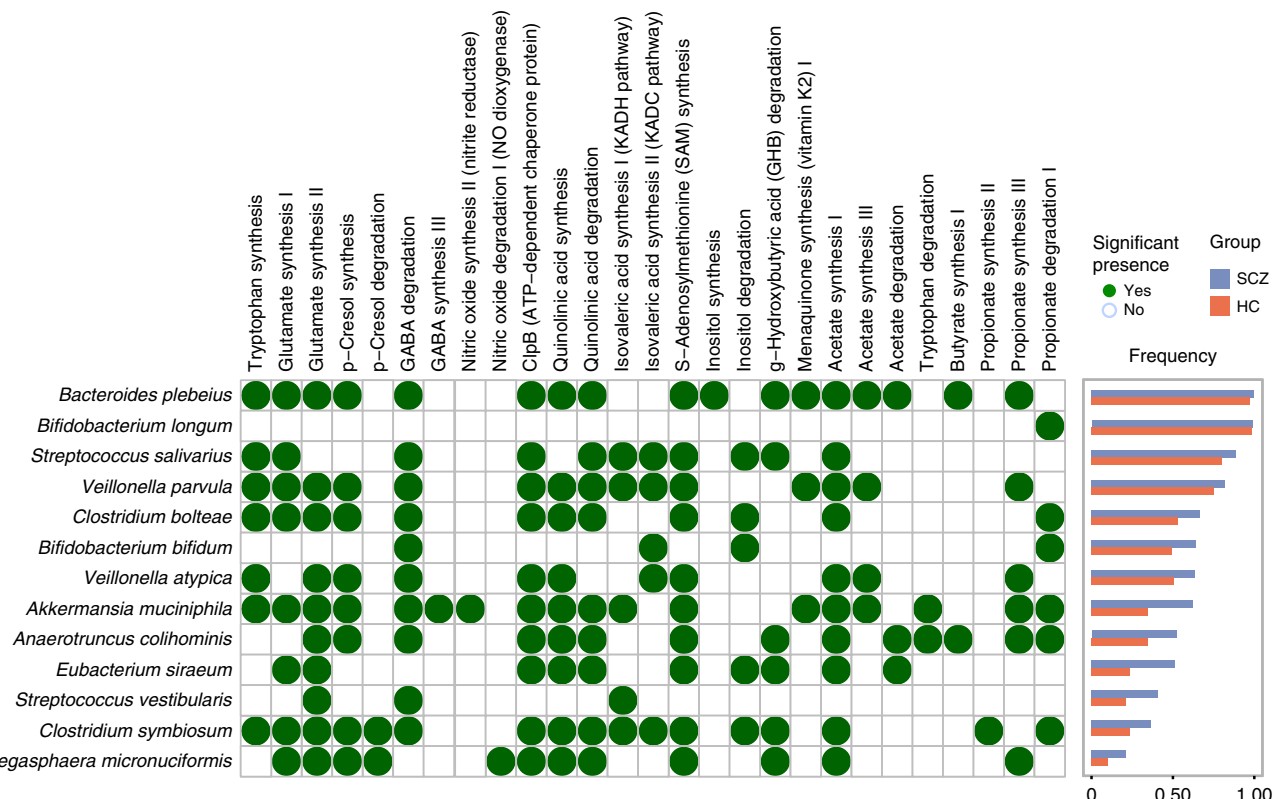

**Fig. 2 The gut-brain modules present in schizophrenia-associated bacterial species.** A green dot indicates a statistically significant association between a gut-brain modules present in schizophrenia-associated bacterial species and a metabolite. No dot represents a non-significant association or a non-existent association. The difference in relation to presence between schizophrenic patients and heathy controls was calculated (Chi-square test, $P < 0.05$). The bar plot shows the frequency of each bacterial species present in schizophrenic patients (SCZ, blue bar) and healthy controls (HC, red bar), respectively. See detailed statistical data in supplementary Source Data file.

and Hochberg method, Fig. 3d). After 3-months of treatment, the abundances of 12 of these 26 mOTUs remained significantly changed compared with the 81 controls ($P < 0.1$, FDR = 0.33, Benjamini and Hochberg method, Supplementary Data 10). Pairwise comparison of all gut mOTUs for treatment effect in the follow-up patients revealed 48 differentially abundant bacterial species ($P < 0.05$ and FDR = 0.420, Paired Wilcoxon rank sum test; Benjamini and Hochberg method, Supplementary Data 11). However, only 5 of the 48 differentially abundant species were included in the 26 mOTUs schizophrenia classifiers. This result suggests that antipsychotic treatment influences the gut microbiota, but does not completely restore the altered microbiota associated with schizophrenia.

**S. vestibularis induced schizophrenia-like behaviors in mice.** *S. vestibularis* contributed to discriminate patients from controls and was associated with serum GABA, tryptophan, KYNA, and the Brief Assessment of Cognition in Schizophrenia (BACS) scores in MATRICS Consensus Cognitive Battery (MCCB) test (Fig. 3 and Supplementary Fig. 2). Moreover, *S. vestibularis*, present in the gut of a number of schizophrenic patients, was predicted to have GBMs related to glutamate synthesis, GABA degradation, and isovaleric acid synthesis (Fig. 2). As some pathogenic species of *Streptococcus* are known to enter the brain[26], and have been implicated in pediatric acute-onset neuropsychiatric syndrome[27,28], we asked if *S. vestibularis* might play a role in the pathophysiology of schizophrenia. Hence, we transplanted *S. vestibularis* ATCC 49124, using oral gavage and drinking water, into C57BL/6 mice after antibiotics-based microbiota depletion

(Supplementary Fig. 4). Another strain of *Streptococcus*, *S. thermophilus* ST12, which is widely present in the human gut, was used as a bacterial control. Behavioral tests were performed to evaluate the effect of *S. vestibularis* transplantation (Fig. 4a). Quantitative polymerase chain reaction (q-PCR) used to quantify the 16S rRNA gene of *S. vestibularis* and *S. thermophilus*, revealed that their concentration increased by 4,164- and 6,183-fold immediately after transplantation and remained at a 31.2- and 58.1-fold increase after the behavioral tests as compared to the control mice (Supplementary Fig. 5). Compared to control mice gavaged with saline or with *S. thermophilus*, the *S. vestibularis*-treated mice exhibited an increase in the total traveled distance and times of rearing during a 30-minute open-field test (Fig. 4b, d). They continued their hyperlocomotion after the 10-minute habituation period and showed no obvious decline in locomotion activity after a period of 30-minutes (Fig. 4c). In the three-chamber social test, the *S. vestibularis* mice displayed obvious deficits in sociability and social novelty, as they were much less sociable and avoided social novelty (Fig. 4e–g). However, in Barnes maze, elevated plus maze, and tail suspension test, the mice transplanted with *S. vestibularis* displayed spatial memory function, depressive state, and anxiety levels similar to either saline or *S. thermophilus*-treated mice (Supplementary Fig. 6). There were no significant changes in body weight, systemic pro-inflammatory cytokines, and endotoxin, and HPA axis hormones between *S. vestibularis*-treated, *S. thermophilus*-treated, and control mice (Supplementary Fig. 7 a–i).

We then compared the transcriptome and neurotransmitter levels between *S. vestibularis*-treated and saline-treated mice in peripheral tissues and brain. *S. vestibularis*-treated mice had

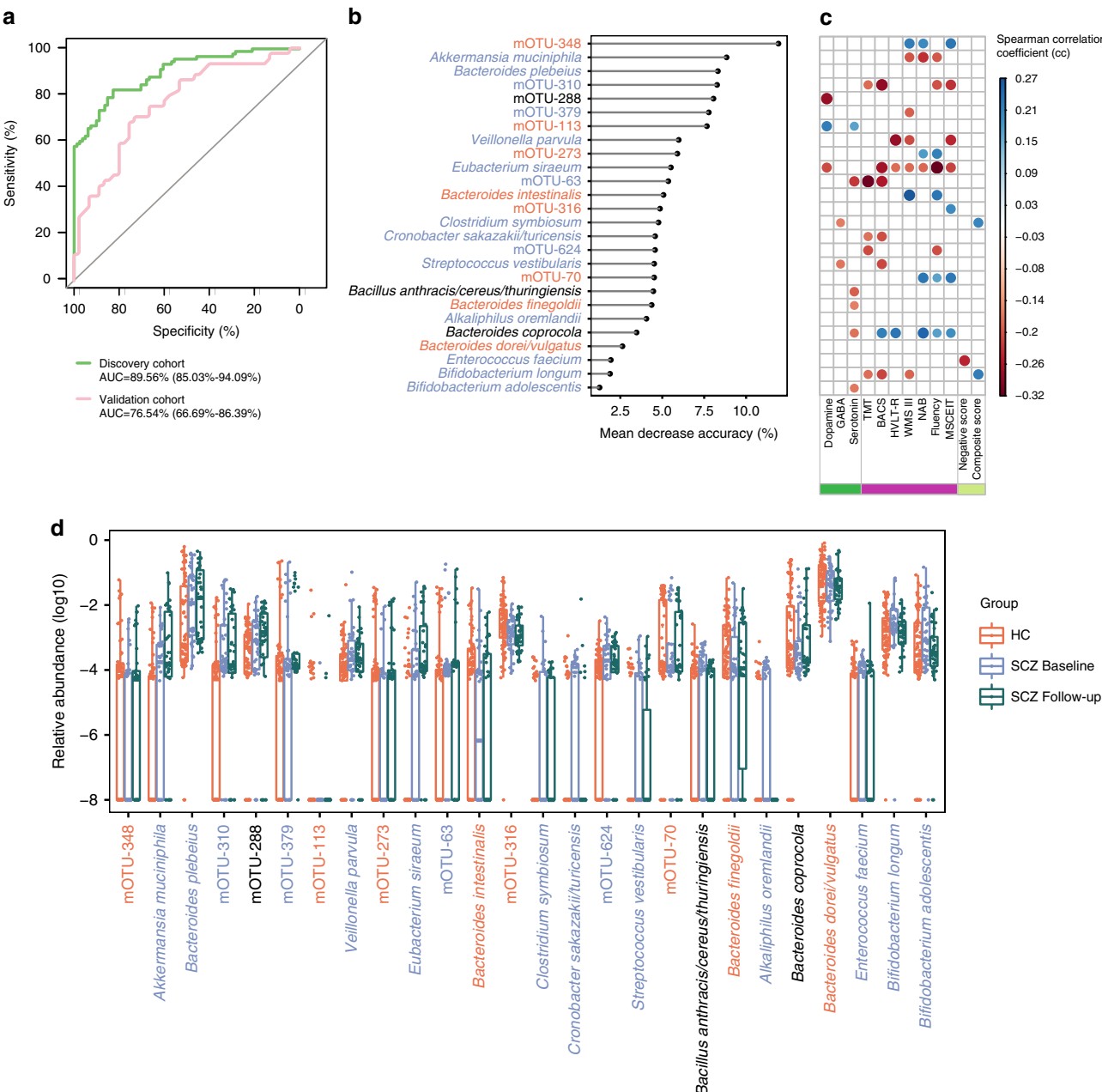

**Fig. 3 Gut microbiome-based discrimination between schizophrenic patients and healthy controls. a** Receiver operating characteristic curve (ROC) according to 171 samples of the discovery set (green line) and 90 independent validation samples (pink line) calculated by cross-validated random forest models. Area under ROC (AUC) and the 95% confidence intervals are also shown. **b** The 26 mOTUs with most weight to discriminate schizophrenic (SCZ) patients and healthy controls (HC) were selected by the cross-validated random forest models. The length of line indicates the contribution of the mOTU to the discriminative model. The color of each mOTU indicates its enrichment in schizophrenic patients (blue) or healthy controls (red) or no significant direction (black), respectively. **c** Spearman's correlation of 26 mOTUs classifiers with three types of neurotransmitter in serum (green), seven types of cognitive function evaluated using the MATRICS Consensus Cognitive Battery (purple), and with the positive score and the negative score of the Positive and Negative Syndrome Scale (light green). Only significant associations are displayed with correlation coefficient (*P*-value < 0.05). **d** The relative abundance (log$_{10}$) of 26 mOTUs classifiers in 90 HCs and 38 SCZ patients at baseline and on a follow-up (3 months later). The dot represents one value from individual participants and boxes represent the median and interquartile ranges (IQRs) between the first and third quartiles; whiskers represent the lowest or highest values within 1.5 times IQR from the first or third quartiles. Outliers are not shown. GABA: 4-aminobutyric acid; TMT: Trail Making Test; BACS: Brief Assessment of Cognition in Schizophrenia; Fluency: Category Fluency in Animal Naming; WMS-III: Wechsler Memory Scale-Third Edition for working memory; HVLT-R: Hopkins Verbal Learning Test-Revised for visual learning; NAB: Neuropsychological Assessment Battery for reasoning and problem solving; MSCEIT: Mayer-Salovey-Caruso Emotional Intelligence Test for social cognition. See detailed statistical data in supplementary Source Data file.

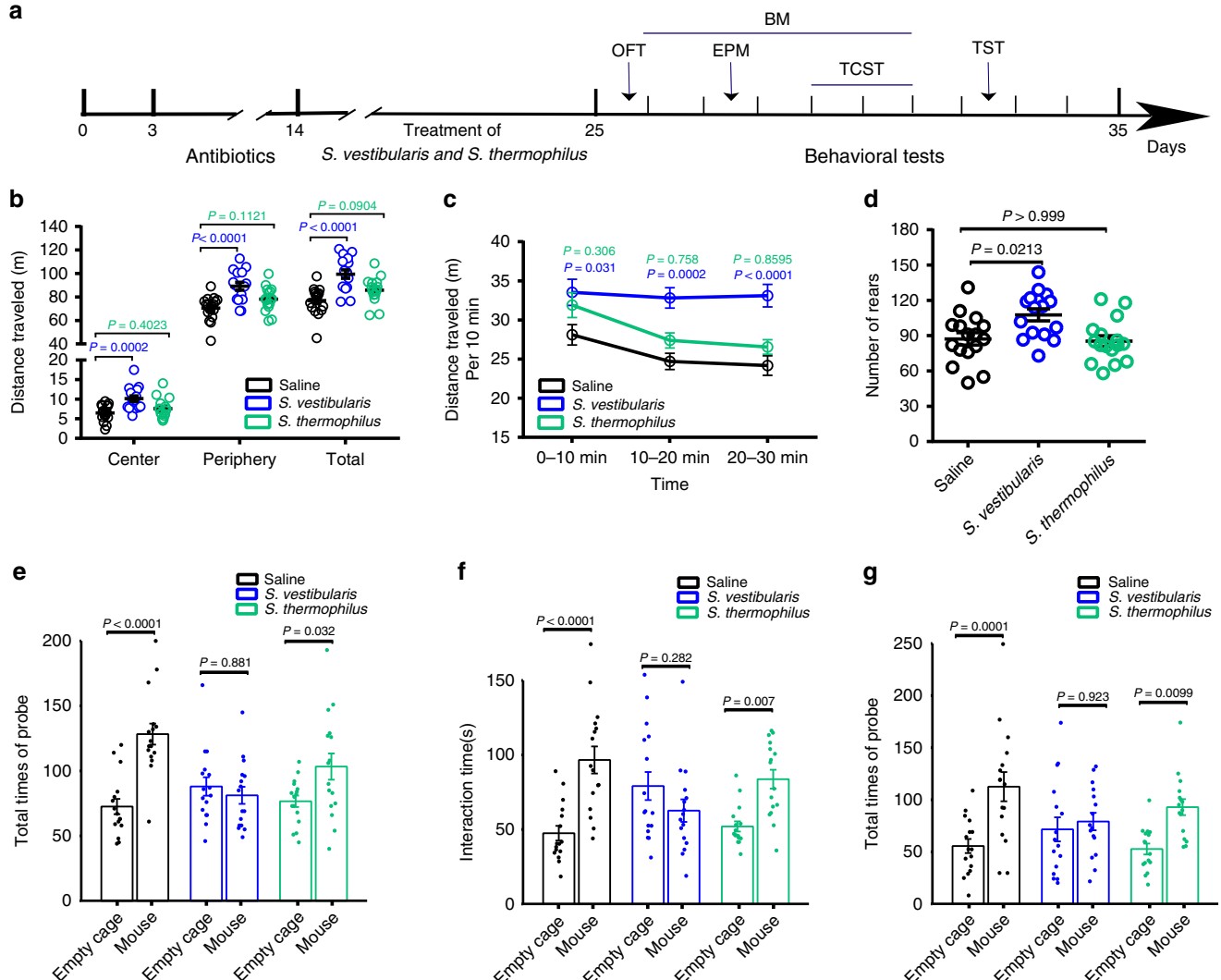

**Fig. 4** ***Streptococcus vestibularis* induces hyperkinetic behavior and impaired social interaction in mice. a** Schematic diagram of bacterial transplantation and behavioral tests. **b** The cumulative distance (meters) in different zones in 30-min Open field test (OFT) in the three groups of mice with oral gavage of *Streptococcus vestibularis*, *S. thermophilus*, and saline, respectively. **c** the cumulative distance (meters) in every 10-minutes time interval of OFT traveled and **d** the number of rearing by *S. vestibularis*-gavaged mice compared to mice gavaged with *S. thermophilus* and control mice. **e–g** three-chamber social test (TCST) comparing sociability of *S. vestibularis*-gavaged mice to that of *S. thermophilus*-gavaged mice and control mice. The results show that *S. thermophilus*-gavaged mice and saline-gavaged mice display obvious sociability, i.e., demonstrate an increase in the number of times probing a mouse (**e**, *P* < 0.0001) and spending longer time interacting with a mouse (**f**, *P* = 0.002) compared to an empty cage, and obvious social novelty, i.e., spending longer time interacting with an unacquainted mouse (new mouse; **g**, *P* = 0.005) in comparison with an acquainted mouse. However, these types of social behaviors were not observed in *S. vestibularis*-gavaged mice (for sociability: **e**, *P* = 0.881; **f**, *P* = 0.282; for social novelty, **g**, *P* = 0.923). The data are representative of two independent experiments and are presented as means ± SEM (*n* = 16 *S. vestibularis*-gavaged mice or *S. thermophiles*-gavaged mice, 17 saline-gavaged mice per independent experiment in OFT; *n* = 16 mice/group/independent experiment in TCST). The circle represents one value from individual mice (**b**, **d**, **e–g**). *P*-values were determined by one-way analysis of variance (ANOVA) (**b**), repeated measure two-way ANOVA followed by Sidak's multiple comparisons test (**c**; Blue *P*: *S. vestibularis*-gavaged versus saline-gavaged mice; green *P*: *S. thermophilus*-gavaged versus saline-gavaged mice), two-sided Kruskal-Wallis test followed by Dunn's multiple comparisons test (**d**), or two-way (ANOVA) followed by Sidak's multiple comparisons test (**e–g**). See detailed statistical data in supplementary Source Data file.

significantly lower levels of dopamine in serum, intestinal contents, and colonic tissue, as well as decreased GABA levels in the intestinal contents immediately after the transplantation, but these effects disappeared 10 days post-transplantation (Supplementary Fig. 8b, e, h, f). Intestinal contents of *S. vestibularis*-treated mice showed increased levels of 5-HT throughout the behavioral tests (Supplementary Fig. 8d). *S. vestibularis* transplantation did not induce obvious inflammatory cell infiltration (Supplementary Fig. 7j), but induced changes in the expression of numerous immune/inflammation-related genes

in the intestine when compared with mice receiving saline gavage (Supplementary Data 12a, b). By gene enrichment analysis, we found that these genes were enriched in cytokine-cytokine receptor interaction, chemokine signaling pathways, leukocyte trans-endothelial migration, complement and coagulation cascades, antigen processing and presentation, intestinal immune network for IgA production, and inflammatory bowel disease (Supplementary Fig. 9a, b). These results suggested that *S. vestibularis* may influence gut immune homeostasis. In the brain, the levels of neurotransmitters were not affected by

transplantation with *S. vestibularis* and only tryptophan decreased in the prefrontal cortex of *S. vestibularis*-treated mice (Supplementary Data 13). However, we observed 354, 540, and 470 significantly differentially expressed gene in the PFC, striatum, and hippocampus, respectively, between *S. vestibularis*-treated and saline-treated mice (Supplementary Data 12c–e). The pathways influenced by these differentially expressed genes include defense responses and immune-regulating pathways, similar to the observed differentially expressed genes in the intestine, as well as peroxisome proliferator-activated receptor signaling pathway, steroid biosynthesis, tyrosine, and tryptophan metabolism (Supplementary Fig. 9c–e).

## Discussion

We here used MWAS to systematically screen for schizophrenia-associated gut microbes and identified a number of schizophrenia-associated GBMs, expanding our insight into functional changes characterizing the gut microbiota of schizophrenic patients. Consistent with the MWAS results and the inferred microbial functions, the schizophrenia-associated bacterium *Streptococcus vestibularis* was shown in mice to have a functional neuroactive potential[20] associated with changes in animal behaviors.

MWAS for the gut microbiome has been controversial due to population differences, diet and medication, and technical differences[29]. With more data from different cohorts, gut microbiome characteristics associated with colorectal cancer have already converged[29,30]. In the present study, a diagnostic model of 26 mOTUs obtained by using a discovery cohort, which included medication-free patients, was well validated in a testing cohort, which included patients taking antipsychotics. We chose to investigate medication-free patients in the discovery cohort in order to eliminate the possible effects of antipsychotics on the gut microbiota to identify gut bacteria possibly involved in the development of schizophrenia. The validation of the initial findings in patients taking antipsychotics demonstrated that abundances of these microbial species are, to a certain extent, independent of antipsychotics. Follow-up analysis also revealed that 22 of the 26 identified mOTUs in the diagnostic model remained the same over a treatment duration of three months. Therefore, most microbial species associated with schizophrenia seem stable and are not sensitive to current antipsychotics.

Analysis of the bacterial V3-V4 region of the 16S rRNA gene regions has a limited resolution in terms of identification of bacterial species[17,22,31]. Current 16S rRNA gene amplicon sequencing generally capture reliable taxonomic classification at the genus level[31]. However, several recent analyses indicate that many taxonomic associations might be presented only at the species level[17,32,33]. Accordingly, most of the schizophrenia-associated microbial species revealed by the MWAS results were not identified in previous studies by using 16S rRNA gene sequencing[14,34,35]. There are more overlaps of findings at the genus level between our findings and the previous studies (Supplementary Data 5b), including six genera in the study of Zheng et al. (*Acidaminococcus, Akkermansia, Alistipes, Citrobacter, Dialister, Veillonella*)[14], one in the study of Schwarz et al. (*Lactobacillus*)[35], one in the study of Shen et al. (*Methanobrevibacter*)[34].

Surprisingly, gut microbiota diversities based on genus level taxonomy and annotated genes were higher in schizophrenic patients than in controls. In accordance with our data, both α diversity and β diversity exhbited an increase in the blood microbiota of schizophrenic patients[36]. The microbes in blood are thought to originate from the gut as well as from the oral cavities[37,38]. Moreover, the increased diversity of the blood

microbiota may be due to the nonspecific overall increased microbial burden in schizophrenia[36], which is supported by our observed increased microbial gene number in the gut of schizophrenic patients. The considerable heterogeneity in the etiology and clinical manifestation of schizophrenia[39,40] may be implicated in such an increase in the microbiota diversity. Another notable feature of the gut microbiota in schizophrenia is the significant enrichment of oral cavity resident bacteria. Increased bacterial translocation due to a leaky gut and innate immune imbalance are both presented in patients with schizophrenia[41,42]. Furthermore, gastrointestinal inflammation due to a dysfunctional immune response to pathogen infections and food antigens is also prevalent in schizophrenia[43,44]. These intestinal pathological conditions may disrupt the mucosal barrier and decrease immune surveillance towards foreign microbes[7], increasing the possibility of observing oral bacteria in the gut.

The composition of human gut microbiota has recently been linked to schizophrenia[14,34,35], but knowledge of individual microbial species is needed to decipher their biological role[45]. We still do not completely understand the functions of most of the schizophrenia-associated microbes identified in the present study or their biological roles in schizophrenia. Intriguingly, some schizophrenia-enriched bacterial species identified in the present study are also over-represented in subjects with metabolic disorders and atherosclerotic cardiovascular diseases[46–49]. Schizophrenic patients are more likely to develop obesity, hyperglycemia/diabetes, hypertension, and cardiovascular disease. Additionally, some of these risks are independent of the effects of antipsychotic administration and healthy lifestyle choices[50–53]. Several prenatal and early-life risk factors are shared by schizophrenia, metabolic disorders, and cardiovascular diseases such as prenatal famine, postnatal growth restriction, the quality of fetal growth, and low birth weight[54–60]. Gut microbes enriched in both schizophrenia and metabolic disorders/cardiovascular disease may account for the increased risk of these comorbidities in schizophrenia. Moreover, new evidence also indicates that metabolic disorders in schizophrenia are not only comorbidities, but also affect pathogenesis, such as the manifestation of negative symptoms[61], cognitive function[62], and brain white matter disruption[63]. Treating metabolic disorders via physical activity and psychosocial and dietary interventions, is also an effective approach to improve the symptoms of schizophrenia[62]. Therefore, manipulation of gut microbes may have double therapeutic potential for both metabolic disorders and schizophrenia.

Using GBMs which were manually curated according to existing knowledge, we identified 27 schizophrenia-associated GBMs, which provides clues as to how the gut microbiota might modulate the pathophysiology of schizophrenia. Among these GBMs, a few well-known molecular entities associated with schizophrenia were covered, such as several types of neuro-transmitters[2–4] and tryptophan metabolites[64,65]. Furthermore, some microbes harboring these GBMs were significantly associated with the serum levels of several neurotransmitters and tryptophan metabolites. Parallel to the present study, our previous animal study demonstrated that transplantation of fecal microbiota from medication-free patients with schizophrenia into specific pathogen-free mice could cause schizophrenia-like behavioral abnormalities and dysregulated kynurenine metabolism[66]. The consistent findings of altered tryptophan-kynurenine metabolism revealed by human serum metabolite analysis, microbiota-based GBM prediction, and mouse studies suggest that this pathway is an important link between schizophrenia and gut microbiota dysbiosis. Of note, transplantation of one bacteria, *S. vestibularis* ATCC 49124, possessing 11 GBMs involved in synthesis and degradation of several types of neurotransmitter induced abnormal behaviors in the recipient mice. The

schizophrenia-enriched *S. vestibularis* contributed to the expression of two types of schizophrenia-relevant behaviors (hyperactivity and impaired social behaviors) in mice, suggesting that GBM prediction is an effective way to screen out potentially functional gut microbes. To the best of our knowledge, this is the first study which aims to determine the functional roles of a single bacterium associated with schizophrenia in disease pathogenesis. Although the biological mechanisms underlying the effects of *S. vestibularis* are still unclear, our data indicate profound influences of this microbe on brain neurotransmitters, and underscore the value of combining MWAS from human cohorts with studies in animals[66]. In conclusion, our study identified a number of schizophrenia-associated bacterial species representing potential microbial targets for future treatment and emphasizes the likely importance of microbial metabolites affecting the development of schizophrenia.

## Methods

**Subject recruitment and clinical assessment**. The present study followed the Declaration of Helsinki and was approved by the Medical Ethics Committee of The First Affiliated Hospital of Xi'an Jiaotong University (TFAHXJTU). It is a publicly registered clinical trial (Identifier: NCT02708316; https://clinicaltrials.gov). Written informed consent was obtained from all participants. Only acutely relapsed schizophrenic (ARSCZ) and first-episode schizophrenic (FESCZ) patients were recruited in this study. Diagnoses were established on the Diagnostic and Statistical Manual of Mental Disorders, fourth Edition (DSM-IV). Healthy controls did not have any mental disorders and were well-matched to the patients on demographic features (Supplementary Data 1 and 9). We interviewed 38 of the schizophrenic patients three months after they joined the project. The antipsychotics dosage, therapeutic response, and side effects were recorded via interview or questionnaire. All assessments were conducted independently by two psychiatrists on the day when the blood samples were collected. Clinical psychopathological symptoms were evaluated by the Positive and Negative Syndrome Scale (PANSS) (Supplementary Data 2)[67]. Cognitive functioning was assessed via the MATRICS Consensus Cognitive Battery (MCCB; the data are showed in Supplementary Data 3)[68,69]. Detailed information on subjects recruitment is presented in the Online methods.

**Shotgun metagenomic sequencing**. Metagenomic shotgun sequencing was performed on Illumina platform for human fecal samples of the discovery cohort (paired end library of 350-bp and 150-bp read length). Adaptor and low- quality reads were discarded from the raw reads, and the remaining reads were filtered in order to eliminate host DNA based on the human reference genome as described previously[70]. Shotgun metagenomic sequencing was performed on the BGISEQ-500 platform for the validation cohort with single-end library and read length of 100 bp. Low-quality reads and host reads were removed[71]. On average, 11.46 Gb and 13.13 Gb of high-quality non-host sequences were obtained per sample in the discovery cohort and the validation cohort, respectively (Supplementary Data 4 and Table 9).

**Taxonomic and functional profiling**. High-quality reads in each sample were aligned to the sequences of a mOTU reference with default parameters[72] and 545 species-level mOTUs were identified. The gene profile was constructed by aligning high-quality reads to the 11.4 M gene catalog[21] by SOAP v2.22 (-m 100 -x 600 -v 8 -p 4 -l 32 -r 1 -M 4 -c 0.95)[73]. The gene abundance was used to calculate alpha diversity, beta diversity. KO assignment was performed using the same procedure as described previously[21]. Putative amino acid sequences were translated from the gene catalog and aligned against the proteins/domains in the KEGG databases (release 79.0, with animal and plant genes removed) using BLASTP (v2.2.24, default parameter except that -e1e -5a6 -b50 -FFm8). Each protein was assigned to a KO by the highest scoring annotated hit(s) containing at least one high-scoring segment pair (HSP) scoring over 60 bits.

**Rarefaction curve analysis**. The rarefaction curve was generated to assess the gene richness in the schizophrenic patients and healthy controls[74]. We performed random subsampling 100 times in the cohort with replacement and estimated the total number of genes that could be identified from a given number of samples.

**α-diversity and β-diversity**. α-diversity (within-sample diversity) was calculated using the Shannon index depending on the gene and mOTU profile[46]. β-diversity (between-sample diversity) was estimated by Bray-Curtis dissimilarity.

**MWAS**. Of the 545 identified mOTUs, we removed mOTUs present in less than 5% of the samples and focus on the remaining 360 mOTUs. The relative abundance of each mOTU was compared between the patients and controls via Wilcoxon rank sum test followed by a Storey's FDR correction. Moreover, mOTUs were correlated with diagnosis via Semi-partial Spearman correlation tests (R package ppcor) adjusting for diet, BMI, age, and gender. The co-occurrence network was visualized using Cytoscape 3.4.0. Pair-wise comparison of all gut mOTUs before and after treatment in the patients with follow-up was conducted via paired Wilcoxon rank sum test with a mutilple testing correction of Benjamini and Hochberg.

Five-fold cross-validation was performed ten times on a random forest model using the mOTUs abundance profiles of the schizophrenic patients and HCs. The test error curves from ten trials of five-fold cross-validation were averaged. We chose the model which minimized the sum of the test error and its standard deviation in the averaged curve[75]. The probability of schizophrenia was calculated using this set of mOTUs and a receiver operating characteristic (ROC) was drawn (R 3.3.2, pROC package). The correlation between gut bacteria abundance and host phenotypes (MCCB, PANSS, neurotransmitter) was calculated by Spearman's correlation. The relationship between "overall diet" and microbial classifier was analyzed by mantel test (ade4 package 1.7–13) revealing that no significant correlation existed between them ($P = 0.78$). Finally, we assessed the possible confounding effects of age, BMI, sex and diet on our random forest model following the procedures of Zeller, et al.[75] (chi-square test, Supplementary Data 8).

**Functional modules predicted from metagenomics**. Bacterial functions were analyzed using two methods, (1) the reporter score and (2) gut–brain modules (GBMs)[20]. The first method depended on the reporter score. Differentially enriched KEGG modules were identified according to their reporter score from the Z-scores of individual KOs. An absolute value of reporter score = 1.96 or higher (95% confidence according to a normal distribution) was used as a detection threshold for modules that differed significantly in abundance[74,76]. The second method involved the use of the 56 GBMs[20] and PanPhlAn[23]. Firstly, we built species profiles and species-specific gene families in our 171 samples, using PanPhlAn (–min_coverage 1 –left_max 1.7 –right_min 0.3). Then, GBM profiles present in the samples were derived by mapping species-specific gene families to the GBM database (blastx, identity 35, score 60, top 1), for each species, and profiling by using the Omixer-RPM version 1.0 (https://github.com/raeslab/omixer-rpm). Finally, species exhibiting significant differences in abundance were identified (chi-square test, $p < 0.05$, FDR = 0.16, Benjamini and Hochberg method, Supplementary Data 7).

**Animals experiment**. Male C57BL/6 J mice were obtained from the Experimental Animal Center of Xi'an Jiaotong University Medical College (five weeks of age; 4–5 per cage). The mice were maintained in a temperature-controlled (21–23 °C) specific pathogen-free level environment with a relative humidity 55 ± 10% and 12/12-h light–dark cycle. Mice were given water and commercial standard feed (produced according to national standard of China for laboratory mouse feed, GB 14924.3) ad libitum. All animal procedures were approved by the Animal Care and Use Committee of Xi'an Jiaotong University. The animal experimental procedure is schematically shown in Supplementary Fig. 4. The body weight was measured, and feces were collected during the single bacterium transplantation experiments and during behavioral tests. The mice were subjected to a series of behavioral task to evaluate the effects of *S. vestibularis* transplantation on locomotor, learning and memory, social behavior, and anxiety and depression level (see Online methods). Mice were killed 24 h after the last behavioral test and peripheral tisses and brain were collected and stored at −80 °C.

Dopamine, gamma-aminobutyric acid, serotonin, kynurenine, kynurenic acid in human serum and mouse serum, as well as the cytokines (IL-1 beta, IFN-γ, and TNF-α) and neurotransmitters (dopamine, 4-aminobutyric acid, 5-hydroxytryptamine) in mouse gut were quantified by Enzyme Linked Immunosorbent Assay (ELISA) according to the manufacturer's instructions (R&D). The detection range, sensitivity, and assay precision are described in Supplementary Data 14. Readings from colonic tissue samples were normalized to total protein content as detected by BCA assay (Sigma-Aldrich, Shanghai, China). Readings from intestinal contents were normalized to total mass of sample. Tryptophan in serum of humans and mice were quantified by liquid chromatography-mass spectrometer. Moreover, tissue concentrations of dopamine (DA), serotonin (5-HT), 4-aminobutyric acid (GABA), and tryptophan in the prefrontal cortex (PFC), the striatum, and the hippocampus of mouse brain were quantified via ultra-high-performance liquid chromatography-tandem mass-spectrometry assay (see Online methods and Supplementary Data 13). Transcriptomes in mice gut and brain were analysis via RNA-seqencing on an Illumina HiSeq 2500 (Illumina, Santiago, CA, USA), at Shanghai Genergy Co., Ltd. (Shanghai, China). Gene expression levels were presented as FPKM- Fragments Per Kilobate of transcript per Million fragments mapped. The enriched biological functions and pathways associated with significantly differentially expressed genes were annotated using gene ontology enrichment analysis and Kyoto Encyclopedia of Genes and Genomes terms.

**Culture and transplantation of *S. vestibularis***. *S. vestibularis* (ATCC 49124) was bought from ATCC (https://www.atcc.org). The strain of *S. thermophilus* ST12 was bought from the Baiobowei company (Beijing, China). The bacteria were cultured according to the provider's instructions, and after washing twice with sterile PBS, they were resuspended in sterile saline at a concentration of $10^8$ and $10^9$ per mL, to be used for oral gavage and drinking water, respectively. Twenty-four hours after

the last gavage of antibiotics, mice were randomly divided into three groups receiving *S. vestibularis*, *S. thermophilus* or saline, respectively. *S. vestibularis* or *S. thermophilus* was provided to the mice via oral gavage (once per day on Day 1–4, 6, 8) and in drinking water for 11 days (drinking water was changed every day). To maintain a high level of the two *Streptococcus* species in the gut of the mice, the mice were offered drinking water with bacteria or saline again for 24 h in the middle of the session of behavioral tests. The schematic of bacterial transplantation and behavioral test is shown in Supplementary Fig. 4. The bacterial abundance in mice gut was measured by real-time quantitative PCR targeting specific DNA sequence in the 16S rRNA gene of the corresponding bacterial genome (Supplementary Data 15).

**Statistical analyses**. Behavioral data are shown as mean ± standard error of the mean (SEM). The comparisons between the three groups (saline, *S. vestibularis* or *S. thermophilus*-treated mice) were carried out using one-way variance analysis (ANOVA) followed by Tukey's multiple comparisons tests or Kruskal-Wallis test and Dunn's multiple comparisons tests. Comparisons regarding repeated weight and the latency to locate escape hole in Barnes maze test were conducted via two-way ANOVA, in accordance with Sidak's multiple comparisons test. Generally, statistical significance is set to 0.05, and Bonferroni correction was applied to the level of significance for multiple testing. All behavioral-related graphs and statistical analyses were generated using GraphPad Prism software, unless otherwise stated. *P*-values, n values, definition of center, and dispersion measurements are indicated in the associated figure legends for each figure.

**Reporting summary**. Further information on research design is available in the Nature Research Reporting Summary linked to this article.

## Data availability
Metagenomic sequencing data for donors and mice samples have been deposited in the CNGB Nucleotide Sequence Archive (CNSA) database under accession identification CNP0000119 and in the European Nucleotide Archive(ENA) database under accession identification code ERP111403. The source data underlying all figures except for those not including statistics are provided as a Source Data file.

## Code availability
The following softwares were used: SOAP v2.22, BLASTP v2.2.24, mOTUs v1, Cytoscape v3.4.0, PanPhlAn v1.2.2.3, Omixer-RPM v1.0, GraphPad Prism v7.0, STAR v2.5.3.a, DESeq2 v1.16.1. The following R packages were used: ppcor 1.0, pROC 1.12.1, randomForest 4.6–14, ade4 1.7–13.

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

## Acknowledgements
This study was supported by the Clinical Research Award of the First Affiliated Hospital of Xi'an Jiaotong University (No. XJTU1AF-CRF-2016-005), Shenzhen Municipal Government of China (DRC-SZ [2015]162), Innovation Team Project of Natural Science Fund of Shanxi Province (2017KCT-20), and Key Program of Natural Science Fund of Shanxi Province (2018ZDXD-SF-036).

## Author Contributions
X.M., H.J., R.G., F.Z., and Z.J. conceived the study. F.Z., Z.Y., J.F., L.Y., B.Z., Q.M., and X.M. performed mice experiments. W.W., Q.M., Y.F., J.F., L.G., Y.D., Y.C., C.C., C.G., and X.M. recruited volunteers and collected samples for the study. W.W., Q.M., Y.F., Z.Y., L.Y., F.Z. and X.M. collected the human fecal and blood samples. Z.Y., L.Y., X.H. and F.Z. analyzed the human serum. R.G., Y.J., Q.W., Q.S. and Y.X. performed bioinformatics analyses. K.K., S.B. and L.M. advised on the mice experiments. F.Z., R.G., H.J., Y.J., and Q.W. interpreted the results and wrote the manuscript with extensive revision performed by L.M., S.B., K.K. and Y.X. L.X., T.Z., X.X., H.Y. and J.W. contributed to the revision and discussion. All authors contributed to the final revision of the manuscript.

## Competing interests
The authors declare no competing interests.
