## [Peer Review File · Nature Communications]

Editorial Note: This manuscript has been previously reviewed at another journal that is not operating a transparent peer review scheme. This document only contains reviewer comments and rebuttal letters for versions considered at Nature Communications .

Reviewers' Comments:

Reviewer #1:

Remarks to the Author:

This substantially improved manuscript is now ok as far as I am concerned from a technical perspective, although it still needs careful and thorough language editing to be suitable for publication.

Reviewer #2:

Remarks to the Author:

The authors have done a valiant job at addressing previous concerns.

I have some remaining issues

The paper would be strengthened by neurobiological correlates of behavioral changes in the mouse studies

The relationship between intestinal GABA and the behavioral changes is very hard to rationalize and is something of a red-herring

The Title is unsatisfactory and alluding to Streptococcus is distracting

Reviewer #4:

Remarks to the Author:

All comments from the reviewer#3 have been addressed.

REVIEWERS' COMMENTS:

Reviewer #1 (Remarks to the Author):

This substantially improved manuscript is now ok as far as I am concerned from a technical perspective, although it still needs careful and thorough language editing to be suitable for publication.

Response: We have performed a thorough editing of the language to make the revised manuscript suitable for publication.

Reviewer #2 (Remarks to the Author):

The authors have done a valiant job at addressing previous concerns. I have some remaining issues The paper would be strengthened by neurobiological correlates of behavioral changes in the mouse studies. The relationship between intestinal GABA and the behavioral changes is very hard to rationalize and is something of a red-herring.

Response: Thank you for your suggestion. We think the neurobiological correlates of behavioral changes in the mouse studies may increase the possibility of overstatement in this manuscript. We agree with you about the improper GABA discussion, we have deleted these languages accordingly.

The Title is unsatisfactory and alluding to Streptococcus is distracting;

Response: We have changed the title following the recommendation of the editor.

The new title is “Metagenome-wide association and preliminary validation of gut microbiome features for schizophrenia”.

Reviewer #4 (Remarks to the Author):

All comments from the reviewer#3 have been addressed.

Response: Thank you very much.